# Integrating Hierarchical Statistical Models and Machine-Learning Algorithms for Ground-Truthing Drone Images of the Vegetation: Taxonomy, Abundance and Population Ecological Models

Christian Damgaard 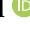

Department of Bioscience, Aarhus University, Vejlsøvej 25, 8600 Silkeborg, Denmark; cfd@bios.au.dk

**Abstract:** In order to fit population ecological models, e.g., plant competition models, to new drone-aided image data, we need to develop statistical models that may take the new type of measurement uncertainty when applying machine-learning algorithms into account and quantify its importance for statistical inferences and ecological predictions. Here, it is proposed to quantify the uncertainty and bias of image predicted plant taxonomy and abundance in a hierarchical statistical model that is linked to ground-truth data obtained by the pin-point method. It is critical that the error rate in the species identification process is minimized when the image data are fitted to the population ecological models, and several avenues for reaching this objective are discussed. The outlined method to statistically model known sources of uncertainty when applying machine-learning algorithms may be relevant for other applied scientific disciplines.

**Keywords:** measurement uncertainty; machine-learning algorithms; plant competition models; hierarchical statistical model

## 1. Introduction

Using drones that record multi-spectral photography and LIDAR, it has now become possible to obtain spatio-temporal ecological data at a fine-scaled resolution. These new data collection possibilities provide a quantum leap compared to earlier methodologies for monitoring ecological processes, e.g., competitive plant growth [1,2]. However, in order to use the drone-aided image data types for modeling plant ecological processes, there is a need to develop statistical models that are especially tailored towards these new image data types [3].

Plant competition is a population ecological process where plant growth is reduced by the presence of neighboring plants. When investigating interspecific interactions in light-open vegetation, the population growth of a species is modeled as a function of the local abundance of other species [4–6]. Previously, plant competitive interactions have been modeled using non-destructive measurements of plant abundance, e.g., using pin-point data, where the vertical density (number of times a plant species is touched by a thin pin) is recorded several times during the growing season in permanent plots. Vertical density is correlated to plant biomass [7,8], and plant growth and interspecific interactions may, consequently, be estimated from repeated pin-point measurements of vertical density [6,9–11]. However, it is now possible to radically upscale the non-destructive measurements of plant abundance by repeated drone-aided recordings of multi-spectral and LIDAR image data of the vegetation. The new image data encompass vast possibilities, but also a new challenge. Compared to pin-point data, which is assembled by persons trained in plant taxonomy, the new image data come without plant taxonomic information or abundance measures.

Currently, image data from drones are being collected in several plant ecological laboratories, and valuable experience on how to recognize plant species is being collected.

It is a natural choice to use machine-learning algorithms for fitting the information from the new image data to observed ground truth of species taxonomy or abundance and, currently, research is focused on how best to use such machine-learning algorithms for predictive purposes in plant ecology [1,12,13].

In the coming years, applied research in predicting the effect of anthropogenic environmental changes on plant community and ecosystem dynamics will surely become more important [14]. Consequently, it is expected that the construction of plant ecological predictions generated by applying machine-learning algorithms on drone-aided image data will be of increasing importance, and it is imperative that such predictions include estimates of prediction uncertainties that are rooted in sampling theory [15–17].

The aim of this study is to outline the principles for using machine-learning algorithms for fitting empirical population ecological models, e.g., competition models, with a known degree of uncertainty. This objective will be met by specifing statistical models that will allow us to quantify the possible bias and uncertainties of species identification and abundance predictions obtained by machine-learning algorithms, so that image data may be used to fit population ecological models with a known degree of uncertainty.

Here, it is proposed to use the confusion matrix of the chosen machine-learning algorithm for quantifying the uncertainty when identifying species taxonomy and integrate a Bayesian hierarchical modeling approach with machine-learning algorithms for quantifying the uncertainty when estimating species abundance. In this study, the proposed general statistical model will be outlined and tentatively specified with suggested relevant statistical distributions. The developed statistical models are needed for fitting population ecological models of plant communities and making quantitative ecological predictions of plant community and ecosystem dynamics, including quantitative assessments of the process or structural uncertainty.

The outline of the manuscript is to present typical non-destructive ground truth data of plant abundance, followed by a brief account of the use of image data and machine-learning algorithms for predicting plant abundance, a detailed proposal of how to model the uncertainty of plant abundance data and how this uncertainty may be integrated into plant ecological models. Finally, the method will be discussed.

## 2. Methods and Models

### 2.1. Pin-Point Data—Vertical Density

In a number of ground-truthing plots at a natural or semi-natural habitat site with light-open vegetation, plant species taxonomic identity and abundance is determined by the non-destructive pin-point method [7,8]. A pin-point frame with $n$ grid points is placed in the vegetation and the position of the frame is recorded using high-accuracy GPS. At each grid point, a thin pin is inserted into the vegetation and the sequence in which different plant species touch the pin is recorded. Such sequence pin-point data allow the determination of several derived plant abundance measures, e.g., cover, top cover and vertical density at the spatial resolution of a single pin or the plot. Furthermore, it is possible to aggregate the species data to higher taxonomic levels or species groups at the pin level.

Depending on the vegetation and the studied ecological question, various measures of plant abundance may be relevant, but here, we will focus on the vertical density at the spatial level of the plot. Importantly, it is assumed that the pin-point measure of vertical density is an unbiased sample of the true, but unknown, vertical density.

### 2.2. Machine-Learning Algorithms of Image Data

A drone was used to record multi-spectral images and LIDAR data of the site with the ground-truthing plots at a resolution that is sufficient to compare the image data with the pin-point data. Using standard image software, [18], a 3D model of the site was constructed and the information of the different bands was summarized at the approximate position of each pin in the pin-point frame. Using supervised machine-learning algorithms, the

taxonomic identity and vertical density of each species was predicted from the image data at the spatial resolution of the plot [1,12].

The species taxonomic identity is predicted from the information in the multi-spectral image bands as well as information on texture etc. [19,20]. In species-rich plant communities, it is to be expected that not all species can be distinguished with sufficient accuracy, and species that cannot be reliably distinguished may be aggregated into a common species group. Since the overall objective of the proposed statistical method is to fit plant population ecological models, it is more important that all plants are accounted for than that each species is identified precisely. Furthermore, when constructing plant population ecological models of species-rich plant communities, it is typically necessary to aggregate plant species into plant species groups or functional groups. In the following, the term species may either mean a single plant species or an aggregated group of plant species.

The vertical density of each species is predicted using the 3D modeling of the vegetation and LIDAR data. It is assumed that the vertical density predicted from the image data may be a biased sample of the true, but unknown, vertical density, and that the direction and magnitude of the bias is species-specific.

### 2.3. Statistical Models

By aggregating species with similar image information, using different auxiliary information, e.g., time series image data, and different supervised machine-learning algorithms, it will be possible to maximize the probability of correct species identification. However, there will always be a non-zero probability of false identification. The probabilities of falsely identifying an entity of vertical density to the wrong species is called a confusion matrix, which is a stochastic matrix, or transition matrix, where each row sums to one. If all species are correctly identified, then the confusion matrix is the identity matrix. The parameters in the confusion matrix are fitted using the data from the ground-truth plots and is, consequently, susceptible to sampling errors; thus, here, it is assumed that each row in the confusion matrix is distributed according to a Dirichlet distribution ($M1$):

$$M1 : \boldsymbol{p}_i \sim Dir(\boldsymbol{\alpha_i}) \tag{1}$$

where $\boldsymbol{p}_i$ is a row vector of $p_{ik}$, which represents the probabilities of classifying species $i$ as species $k$, and $\boldsymbol{\alpha_i}$ is a row vector of $\alpha_{ik}$, which represents the number of times species $i$ is categorized as species $k$ by the supervised machine-learning algorithm [21].

The hierarchical model for determining the uncertainty of the vertical density measured by the drone images is outlined in Figure 1. The true, but unknown, vertical density of species $i$ at plot $j$ is denoted $x_{ij}$. The pin-point vertical density of species $i$ at plot $j$ observed by the pin-point method is denoted $y_{ij}$, and assumed to be distributed according to a generalized Poisson distribution ($M2$) with mean parameter $x_{ij}$ and a species-specific scale parameter $\rho_i$ [9,22]. The predicted vertical densities from the image data at the level of the plot are denoted $m_{ij}$ and assumed to be distributed according to a reparametrized gamma distribution ($M3$) with mean $x_{ij} + \tau_i x_{ij}$, where $\tau_i$ is a species-specific bias parameter and $v_i$ is a species-specific scale parameter:

$$M2 : y_{ij} \sim GP\big(x_{ij}, \rho_i\big) \tag{2}$$

$$M3 : m_{ij} \sim Gamma\big(x_{ij} + \tau_i x_{ij}, v_i\big) \tag{3}$$

The idea is now to fit the measurement equations $M1$ and $M3$ to the information in the ground-truthing plots and keep these fitted measurement equations fixed when fitting the plant population ecological models to the image data of the whole site.

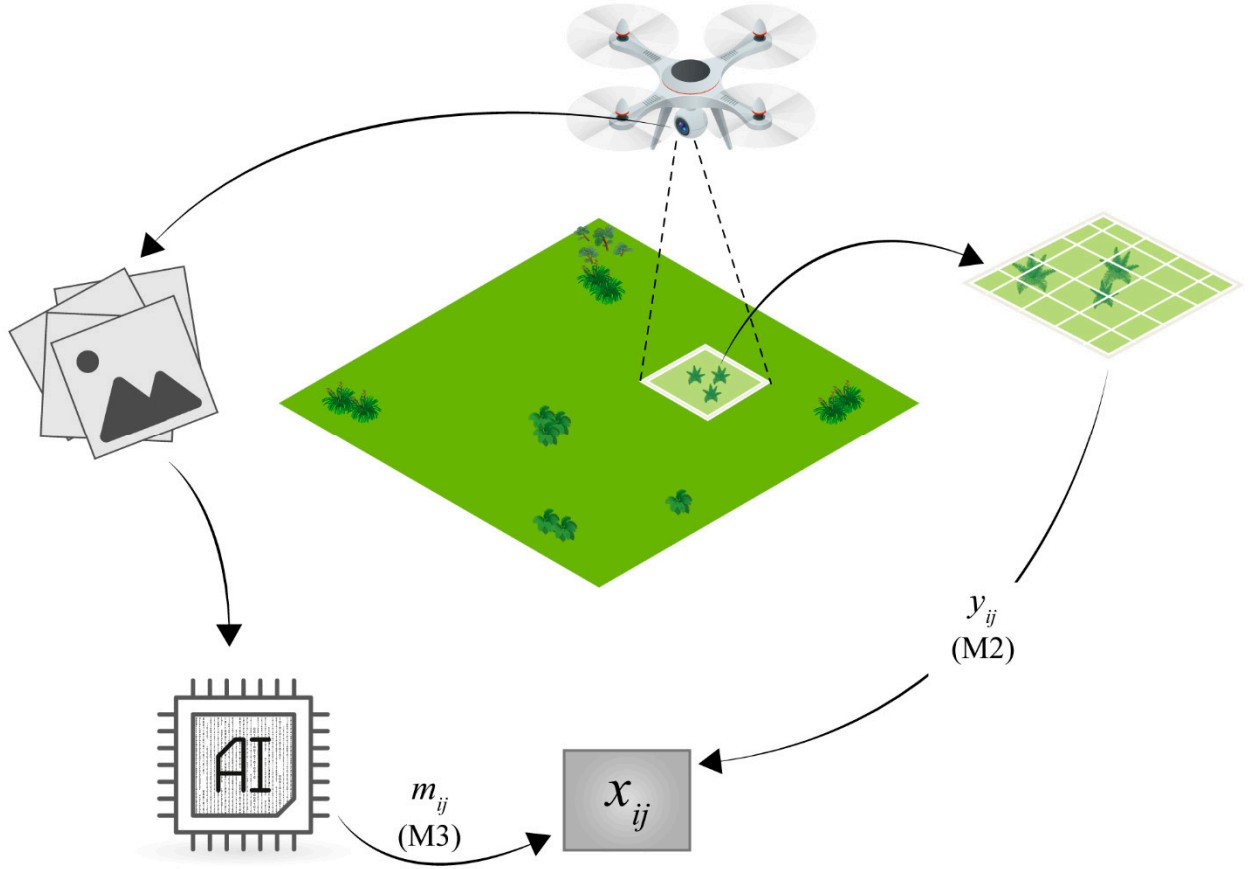

**Figure 1.** Outline of the hierarchical model for determining the uncertainty of the vertical density measured by the drone images. The true, but unknown, vertical density of species $i$ at plot $j$ in ground-truthing plot $j$ is modeled by the latent variable $x_{ij}$. The posterior distribution of the latent variable is calculated using both (i) the vertical density predicted from the information from the drone images using machine-learning algorithms ($m_{ij}$) that are modeled using *M*3, and (ii) the vertical density measured by the pin-point method ($y_{ij}$) that is modeled using *M*2.

### 2.4. Population Ecological Modeling Using Image Data

The ultimate aim of the statistical models is to be able to fit plant population ecological models, e.g., competition modes, to time-series image data with a known degree of uncertainty. Following a discrete Lotka-Volterra competition model and earlier population ecological modeling studies, where interspecific interactions are modeled using pin-point abundance data [6,9–11], the following general species interaction modeling framework may be followed:

$$x_{i,t+1} = f_i(x_{i,t}) \sum_j Exp(-c_{ij} \, x_{j,t}) \qquad (4)$$

where $f_i$ is a species-specific growth function in the absence of interspecific interactions and $c_{ij}$ measure the competitive effect of species $j$ on the growth of species $i$.

The population ecological model (Equation (4)) may now be applied on a selected "vegetation plot" $l$ that has the same size as the ground-truthing plots, but where only image data are available. The model (Equation (4)) is the process equation in a hierarchical model, where the measurement equation of the true, but unknown, vertical density of species $i$ in a selected "plot" $l$ at time $t$, $x_{il,t}$ is specified by the predicted vertical density $m_{il,t}$ and the fitted *M*3 (Figure 2).

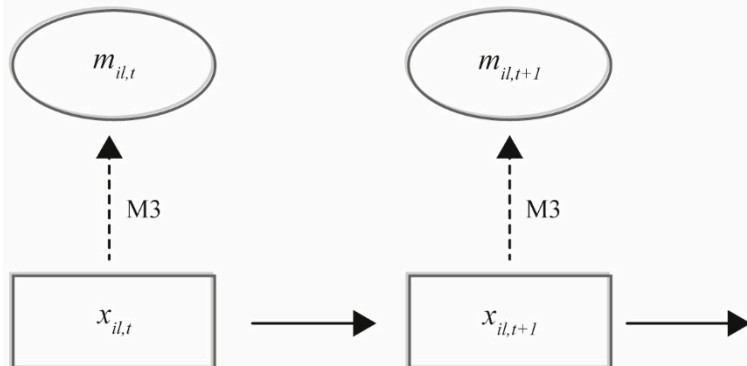

**Figure 2.** Hierarchical population ecological model fitted to image data from a selected "vegetation plot" *l* that has the same size as the ground-truthing plots, but where only image data are available. The true, but unknown, vertical density of species *i* at time *t* is modeled by the latent variable $x_{il,t}$ and the solid arrows are the process equation (Equation (4)). The dashed arrows are the fitted measurement equations (*M3*) that link the vertical density predicted from the information from the drone images ($m_{ij}$) to the latent variables.

The output of running the selected machine-learning algorithms on the image data of plot *l* is a vector, $m_l$, where each element in the vector contains the predicted species identity and the corresponding predicted vertical density of that species in the plot. However, the species identity is determined with some uncertainty from the image data, and this uncertainty needs to be included in the uncertainty of the population ecological modeling. This uncertainty is proposed to be included when fitting the model using a numerical MCMC procedure by drawing $m_l^d$ during the model fitting procedure according to $m_l$ and the fitted *M1*. More specifically, for each entity of vertical density in $m_l$, a new species identity is randomly drawn using the fitted *M1*, and the resulting vertical densities are collected by their drawn species identity into the matrix $m_l^d$. The frequency of drawing a new $m_l^d$ may be set to every 100th MCMC iteration, but the sensitivity of this frequency setting to the overall convergence properties of the MCMC must be checked by visual inspection of the sampling chains.

## 3. Discussion

Generally, when making ecological predictions, it is important that the measurement and sampling uncertainty is taken into account, e.g., by the explicit modeling of the error due to measurement and sampling in a hierarchical model; otherwise, the predictions may be biased due to regression dilution [11]. More specifically, such prediction biases have been demonstrated when omitting measurement errors in plant competition models [11].

In the outlined hierarchical modeling framework, it is demonstrated how measurement and sampling uncertainty may be modeled when fitting population ecological models to drone image data. The chosen statistical distributions (*M1*, *M2* and *M3*) are natural choices for modeling the statistical uncertainty of the different stochastic processes and, except for *M3*, they have been applied in a number of empirical studies [6,9–11]. However, the outlined modeling concept is general, and alternative specifications of the suggested statistical distributions may be relevant in other cases. For example, the bias correction in *M3* is suggested to be proportional to the vertical density, but if more detailed information on the bias is available, then this information should, of course, be used to specify *M3* [15].

The reason for choosing vertical density obtained by the pin-point method as the measure of plant abundance in the ground-truthing plots is three-fold: (i) the vertical density is a non-destructive method for measuring plant abundance that has been shown to be correlated with plant biomass [7], (ii) the vertical density measure has previously been shown to be useful for fitting plant population ecological models [6,9–11] and (iii) it is possible to aggregate the abundance of single species into the abundance of species groups or plant functional types [23]. However, other measures of plant abundance with

similar characteristics may be used instead, after which the statistical distribution used in M2 should be modified accordingly.

Generally, it is important that abundance measures allow for the aggregation of abundances across species groups, e.g., counts of individuals, biomass or vertical density. In species-rich communities, it will not be practically possible, or even desirable, to construct dynamic population ecological models where all species are accounted for individually. Instead, it is important to construct taxonomic or ecologically meaningful species groups that allow the results of population ecological models to be generalized across sites [24]. This necessity to group species may be compared to the plant trait-based approach of summarizing the ecological functions of local plant communities by the mean and variances of selected plant traits [23,25].

It is critical that the error rate in the species identification process is minimized when the image data are fitted to the population ecological models. In order to meet this requirement, a number of actions can be applied: (i) use time-series image data to identify species-specific changes in the image data, (ii) aggregate species with similar characteristics in the image data into a species group and (iii) only select plots with species groups that are clearly distinct in the image data for population ecological modeling. Regarding the later suggestion, note that in the population ecological modeling of plants, competitive growth it is not necessary to include all plots or a random selection of plots in the fitting process. Instead, it is a valid approach to select plots and model competitive interactions where species of particular interest are locally coexisting [9].

In this study, we have focused on how to model the uncertainty when fitting population ecological models to drone image data, for example, when studying the effect of environmental gradients (e.g., nitrogen deposition, precipitation, grazing intensity and herbicide drift) on plant population growth. The general outcome of such an analysis will be the joint posterior probability distribution of the model parameters, which may be used to test hypotheses and make quantitative predictions on the effect of the studied environmental gradient [15]. Such results may be important for predicting the effect of anthropogenic environmental changes on plant community and ecosystem dynamics, and for recommending mediation strategies of possible negative effects of such changes.

**Funding:** The work is supported by an AnaEE grant.

**Conflicts of Interest:** The author declares that he has no conflict of interest.

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
