# Peer review of "Integrating Hierarchical Statistical Models and Machine-Learning Algorithms for Ground-Truthing Drone Images of the Vegetation: Taxonomy, Abundance and Population Ecological Models"

_remotesensing, doi:10.3390/rs13061161_

Round 1

Reviewer 1 Report

The paper lacks novelty. Moreover, it is not structured in a correct way.

Author Response

Reviewer 1:

The paper lacks novelty.

The modelling of sampling and measurement uncertainty in population ecological models has just started to receive attention in the ecological literature, and I therefore find the discussion of this topic both timely and highly relevant. I have now rewritten parts of the Discussion to stress the importance of including sampling and measurement uncertainty in ecological models and added two recent references.

Moreover, it is not structured in a correct way.

The paragraphs describing the aim of the study, and the account of how the paper is structured has been modified. The Discussion has been restructured.

Please try to be more specific in what is wrong with the structure and I will try to correct it.

Reviewer 2 Report

Dear author,

I had the pleasure to review your work. The work is good, however, it is conceptual and theoretical. The equations should be validated by taking some sample data.

It would be good if you take some sample data and try validating your method otherwise this work should go under conceptual/theoretical work.

Thank you!

Author Response

Reviewer 2:

I had the pleasure to review your work. The work is good, however, it is conceptual and theoretical. The equations should be validated by taking some sample data.

The measurement equations that models the ground truth data has been validated in the cited literature.

It would be good if you take some sample data and try validating your method otherwise this work should go under conceptual/theoretical work.

It is true that the work is conceptual/theoretical. Is there a special classification for conceptual/theoretical papers that I am not aware of?